# Relative tumor volume has prognostic relevance in canine sinonasal tumors treated with radiation therapy: A retrospective study

**Felicitas Czichon**[1], **Carla Rohrer Bley**[1], **Valeria Meier**[1,2]*

**1** Vetsuisse Faculty, Division of Radiation Oncology, Department for Small Animals, University of Zurich, Zurich, Switzerland, **2** Department of Physics, University of Zurich, Zurich, Switzerland

* vmeier@vetclinics.uzh.ch

**Data Availability Statement:** The data that support the findings of this study are openly available in Harvard Dataverse at [https://dataverse.harvard.

## Abstract

Tumor volume is controversially discussed as a prognostic factor in dogs treated with radiation therapy for sinonasal tumors. Dogs' body sizes vary widely and relative rather than absolute tumor volume might provide better prognostic information. Our hypothesis was that relative rather than absolute tumor volume (gross tumor volume, GTV) influences time to progression (TTP) and that a larger tumor volume is correlated with a higher tumor stage. We retrospectively investigated possible correlations of initial GTV to weight, body surface area (BSA), nasal cavity size and the tumor stage in 49 dogs with sinonasal tumors. Here, also presumed sinonasal tumors, esthesioneuroblastomas and histologically benign tumors were included. The possible impact of absolute and relative GTV on response and outcome were assessed according to imaging findings in 34 dogs with available follow-up computed tomographies (CTs) after definitive-intent radiation therapy with either a regular (10x4.2 Gy) or a simultaneously- integrated boost protocol (SIB; GTV boosted to 10x4.83 Gy). In contrast to absolute GTV (p<0.001), the relative GTVs were not correlated with dogs' body sizes. Absolute GTV, GTV relative to weight and BSA were not associated with TTP based on CT imaging. However, GTV relative to nasal cavity showed a prognostic influence with a hazard ratio of 10.97 (95%CI:1.25–96.06). When looking at GTV relative to nasal cavity, stage 3 and 4 tumors were significantly larger than stage 1 and 2 tumors (p = 0.005). Our results suggest that GTV *relative to nasal cavity* could be prognostic for TTP and a larger tumor volume relative to nasal cavity is correlated with a higher tumor stage.

## Introduction

Tumor volume remains poorly described as a prognostic factor in dogs treated with radiation therapy for sinonasal tumors. The few reports that described tumor volume and its association with outcome showed contradictory results. On the one hand, neither relative tumor regression (decrease in tumor volume in relation to nasal cavity), nor absolute tumor volume and percentage volume change at 1.5–3 months after radiation therapy were associated with overall survival or progression-free interval [1,2]. On the other hand, both, tumor volume and degree

edu/dataset.xhtml?persistentId=doi:10.7910/DVN/
UUHLPE].

**Funding:** This work was supported by the Swiss National Science Foundation (SNSF), http://www.snf.ch/en, grant number: 320030-182490 (PI: Carla Rohrer Bley). The funders had no role in study design, data collection and analysis, decision to publish, or preparation of the manuscript.

**Competing interests:** The authors have declared that no competing interests exist.

of tumor response to radiation therapy were found prognostic for progression-free survival and overall survival [3]. In this latter study, initial tumor volume below the median and complete response yielded prolonged outcome.

In contrast, tumor stage was evaluated in several studies. Nevertheless, the importance of tumor stage as prognostic parameter remains controversial. For example, some reports consider higher tumor stage to be prognostically unfavourable [4–6], but in other studies tumor stage was found not to be significantly associated with progression- free interval, progression-free survival or overall survival [2,3,7].

Recently, volume response to radiation therapy at different time points was described more closely using either the longest diameter with the response evaluation criteria in solid tumors (RECIST) or direct 3-dimensional (3D) volume measurement. 3D delineation of the volume was superior to measurement of longest diameter because volume measurement did not underestimate tumor volume compared to the determination of longest diameter [2].

Thus, tumor volume measurement seems to be important when evaluating response and outcome. However, dogs' body sizes vary widely. A tumor of a certain volume might obstruct the nasal airway passage completely in a Chihuahua but occupies only a small part of the nasal cavity of a Great Dane. Thus, absolute tumor volume might not best represent the individual dog's tumor size and its implications. Relative tumor volume might be better suited for different dog sizes and may shed further light on its ability to provide prognostic information. Further insight into both, tumor volume and stage, their possible correlation and influence on outcome are therefore warranted.

In this study, we retrospectively investigated initial tumor volume, correlated it with tumor stage and looked at the response to treatment with either a regular (10x4.2 Gy) or a simultaneously- integrated boost protocol (SIB; GTV boosted to 10x4.83 Gy). First, we evaluated a possible correlation of initial tumor volume to weight, body surface area (BSA), nasal cavity size and the tumor stage in a cohort of dogs with sinonasal tumors. We then assessed response and outcome on follow-up computed tomographies (CTs). As a volume parameter, we investigated not only the absolute GTV, but also the tumor volumes corrected in relation to the dogs' weight, BSA and nasal cavity size.

Our aim was to further investigate tumor volume as a prognostic factor by taking different dog sizes into account, as well as correlating tumor volume with tumor stage. We hypothesized that relative rather than absolute GTV influences time to progression (TTP) and that a larger tumor volume is correlated with a higher tumor stage.

## Material and methods

### Case selection

Datasets from a previous study were evaluated for this single institution retrospective cohort study [8]. The mentioned study was non-randomized and included dogs between June 2014 and May 2020 with non-lymphomatous, sinonasal tumors treated with either our regular radiation therapy protocol (10x4.2 Gy) or with a simultaneously- integrated boost protocol (SIB; GTV boosted to 10x4.83 Gy) and was approved by the Animal Ethics Council of the Canton of Zurich, Switzerland (permit number: ZH075/17). All subjects were client-owned dogs that underwent definitive-intent, daily (Monday-Friday) dynamic intensity-modulated radiation therapy (IMRT) at the Division of Radiation Oncology, Vetsuisse Faculty, University of Zurich.

In part one of our study, we assessed correlations of the GTVs to different dog sizes: namely, weight, BSA and nasal cavity. This led to different relative GTVs. We then evaluated a possible correlation of initial tumor volume to the tumor stage. Inclusion criteria for part one

was a dog with a cytologically or histologically diagnosed sinonasal tumor. Dogs with unknown tumor histology or benign tumors were also included, either because of highly suspicious CT characteristics for a malignant nasal tumor or because of severe clinical nasal signs incompatible with acceptable quality of life. Not only including malignant sinonasal tumors seemed reasonable, as the goal of part one of the study was correlation of different tumor volumes to different dog sizes.

In part two of this study, we assessed response and outcome on a subset of irradiated dogs as described above and with available follow-up CTs. Because response and outcome were only evaluated based on imaging findings, inclusion criteria for part two were the same as for part one but required a re-check CT scan at any time point after the end of radiation therapy. This means, cases were excluded for part two, if they didn't have a follow- up CT.

## Patient and tumor characteristics

Information from medical reports consisted of breed, age, sex, weight, tumor type, tumor stage, GTV, clinical target volume (CTV) and planning target volume (PTV). Stage based on the modified Adams tumor staging system [4] was determined according to the description in the original CT report (written by a board-certified radiologist).

## Treatment set-up, contouring and planning

Treatment set-up (rigid bite-block system and vacuum cushion), radiation equipment (Varian Clinac iX 6MV linear accelerator and ECLIPSE planning software version 10.0.28 or 15.1.25 (Varian Oncology Systems, Palo Alto, USA)), contouring of target volumes and organs at risk, and treatment planning was performed as described by Meier at al. [8]. In short, the GTV included the contrast enhancing mass seen in the co-registered contrast-enhanced CT images, as well as the lytic bone visualized in the bone window. The sinonasal cavity was included in the CTV, in case of GTV extension in that part or if it was filled with fluid. Additional craniocaudal CTV extension in the sinonasal cavity or nasopharynx was 1.5 cm for carcinomas and benign tumors and 2 cm for sarcomas or unknown tumor histotypes. A 2 mm extension from the CTV led to the PTV. Contouring was performed by the responsible board-certified radiation oncologist (CRB or VM) as part of regular treatment.

Intended dose coverage was as follows: 98% of the PTV had to receive 95% or more of the prescribed dose of 42 Gy. For the SIB protocol, 98% of the GTV had to receive 95% or more of the prescribed SIB dose of 48.3 Gy. Dose maxima up to 107% of the respective dose levels in any volume and dose maxima of >110% in a small volume (<2%) within the PTV were allowed. Dose to target volumes as recommended by Rohrer Bley et al. and Keyerleber et al. [9,10] was described in the original study [8].

The nasal cavity (NC) was defined according to an anatomical textbook and contoured retrospectively in all dogs starting rostrally from the nostrils to the ventral nasal meatus caudally, including the nasal septum and frontal sinus [11]. Contouring was performed by a veterinarian (FC) and checked by a board-certified radiation oncologist (VM). The BSA was computed using the formula according to Pereira et al.: BSA (cm$^2$) = 10.1 x weight (grams)$^{2/3}$ [12]. The relative GTV was calculated as the ratio between the dogs' GTV and the respective dog size and was defined in terms of the following three patient-specific factors: weight (GTVrel_W), BSA (GTVrel_BSA) and nasal cavity (GTVrel_NC).

## Follow-up CT, response assessment and outcome

All follow-up CTs (acquired after contrast agent administration) were imported into the treatment planning system and registered with the original radiation planning CT. A copy of the

original GTV was made, manually adjusted to the smaller or bigger tumor volume of each re-check CT and compared with the original GTV or the smallest re-check GTV. The response to treatment was assessed for each follow-up CT using the volume response determination described by Nell et al. (2020) with GTV percentage volume change based on World Health Organization (WHO) guidelines and classified as follows: complete response (CR) if no residual tumor was visible, partial response (PR) in case of >50% decrease in volume, progressive disease (PD) if >25% increase in volume and stable disease (SD) as volumetric change insufficient to qualify as response or progression [2]. The best response was defined as the best tumor response confirmed by imaging recorded across all time points [13]. If only one re-check CT was available at the time of tumor progression, best response was progressive disease. Tumor progression was therefore defined as an increase in tumor size according to computed tomography as described by Nell et al. [2] (and not according to new/ recurrent clinical nasal signs). Information regarding outcome was based on previous data collected from medical records or phone calls to the referring veterinarian or the owner. Re-check CTs were recommended every 6 months after the end of radiation therapy or in case of newly occurring nasal signs (but could vary due to owner preference).

## Statistical methods

Data were coded in Excel (Microsoft® Excel® for Microsoft 365 MSO) and analyzed by using SPSS (IBM SPSS Statistics, Version 27) and R (http://www.R-project.org/).

For the correlation analysis, we used the cohort of patients from part one. For the evaluation of tumor response and outcome, we used the subset of dogs with follow-up CTs from part two. Descriptive statistics were reported with mean (standard deviation) for normally distributed data and median (95% confidence interval (95% CI) or interquartile range (IQR)) for non-normally distributed data. Statistical analysis testing for normal distribution and correlations were performed using Shapiro-Wilk test and Pearson correlation. A two-sided independent t-test for normally distributed data, Wilcoxon test for non-normally distributed data and log-rank for survival analysis test were performed to compare the regular group with the SIB group. Kruskal-Wallis one-way analysis of variance test was used to evaluate correlations between GTVs (absolute and relative) and tumor stages. Pair-wise Wilcoxon tests were used to assess differences between GTVs and tumor stage. Tumor progression was evaluated by means of imaging and the three-dimensional response evaluation by Nell et al. (2020). If no imaging was available but the dog showed clinical nasal signs compatible with progressive disease, the dog was considered to show progressive disease as well. TTP was defined as the interval between radiation therapy start and disease progression according to the CT images [2] or clear clinical nasal signs. Dogs were censored if they died of a tumor-unrelated cause or if they were alive and free of progression at the time of data evaluation. Kaplan-Meier plotting was used for visualization of TTP. Cox regression was used to determine the effect of tumor volume (absolute versus relative) on TTP. Results with a p-value <0.05 were considered as statistically significant.

## Results

### Patient and tumor characteristics

Forty-nine dogs with sinonasal tumors treated with radiation therapy were used for analysis of correlations (part one). Thirty-four dogs had a re-check CT scan at time points after the end of radiation therapy and therefore met the inclusion criteria for evaluating tumor response and outcome (part two). Eighteen dogs with follow-up CT were irradiated with the regular protocol and sixteen with the simultaneously- integrated boost protocol. Patient and tumor characteristics are summarized in Tables 1 and S1. In all three dogs with unknown tumor histology, a

**Table 1.  Patient and tumor characteristics for part two (n = 34).**

| | Regular protocol (n = 18) | SIB protocol (n = 16) | Total (n = 34) | P-value |
|---|---|---|---|---|
| **Age*** (years) mean (±SD) | 10.5 (±2.7) | 9.8 (±2.1) | 10.2 (±2.4) | 0.42 |
| **Sex** n = (%) Female Female spayed Male Male castrated | 3 (16.7%) 5 (27.8%) 4 (22.2%) 6 (33.3%) | 2 (12.5%) 8 (50%) 1 (6.3%) 5 (31.3%) | 5 (14.7%) 13 (38.2%) 5 (14.7%) 11 (32.4%) | 0.415 |
| **Weight*** (kg) mean (±SD) | 22.6 (±11.4) | 22.6 (±9.9) | 22.6 (±10.6) | 1.00 |
| **BSA*** ($m^2$) mean (±SD) | 0.8 (±0.3) | 0.8 (±0.2) | 0.8 (±0.3) | 0.93 |
| **Nasal Cavity*** ($cm^3$) mean (±SD) | 103.6 (±59.8) | 108.3 (±50.2) | 105.8 (±54.7) | 0.81 |
| **Tumor type** n = (%) Carcinoma Sarcoma Benign** Unknown Esthesioneuroblastoma | 9 (50%) 7 (38.9%) 1 (5.6%) 1 (5.6%) 0 (0.0%) | 6 (37.5%) 5 (31.3%) 1 (6.3%) 2 (12.5%) 2 (12.5%) | 15 (44.1%) 12 (35.3%) 2 (5.9%) 3 (8.8%) 2 (5.9%) | 0.53 |
| **Tumor stage** n = (%) Stage 1 Stage 2 Stage 3 Stage 4 | 2 (11.1%) 5 (27.8%) 3 (16.7%) 8 (44.4%) | 4 (25%) 2 (12.5%) 3 (18.8%) 7 (43.8%) | 6 (17.6%) 7 (20.6%) 6 (17.6%) 15 (44.1%) | 0.59 |
| **GTV abs** ($cm^3$) median [IQR] | 36.60 [20.97, 65.25] | 26.15 [11.87, 55.10] | 31.80 [15.15, 57.10] | 0.47 |
| **GTV rel_W** median [IQR] | 1.59 [1.11, 2.60] | 1.30 [0.98, 2.03] | 1.43 [1.11, 2.11] | 0.33 |
| **GTV rel_BSA** median [IQR] | 42.33 [33.59, 69.14] | 39.90 [22.64, 57.51] | 39.90 [25.90, 67.22] | 0.285 |
| **GTV rel_NC** median [IQR] | 0.35 [0.27, 0.51] | 0.35 [0.15, 0.44] | 0.35 [0.26, 0.46] | 0.285 |

GTVabs: Absolute tumor volume; GTVrel_W: Tumor volume relative to weight; GTVrel_BSA: Tumor volume relative to body surface area; GTVrel_NC: Tumor volume relative to nasal cavity; IQR: Interquartile range; SD: Standard deviation; *: Data was normally distributed according to Shapiro- Wilk test and is therefore presented as mean (±SD).

**: Angiofibroma, extensive polyps.

biopsy was attempted but was non-diagnostic. The dogs were included because CT characteristics were highly suspicious for a malignant nasal tumor (space-occupying soft tissue attenuating mass with aggressive bone lysis). The two dogs with benign tumors were included because they showed large tumor extension and complete obstruction of the nasal pathways incompatible with acceptable quality of life (euthanasia was considered by both owners). Histopathological diagnosis was an angiofibroma in one dog (local progression 220 days after the end of radiation therapy) and extensive polyp formation in the other dog (euthanized 500 days after the end of radiation therapy due to breathing difficulties).

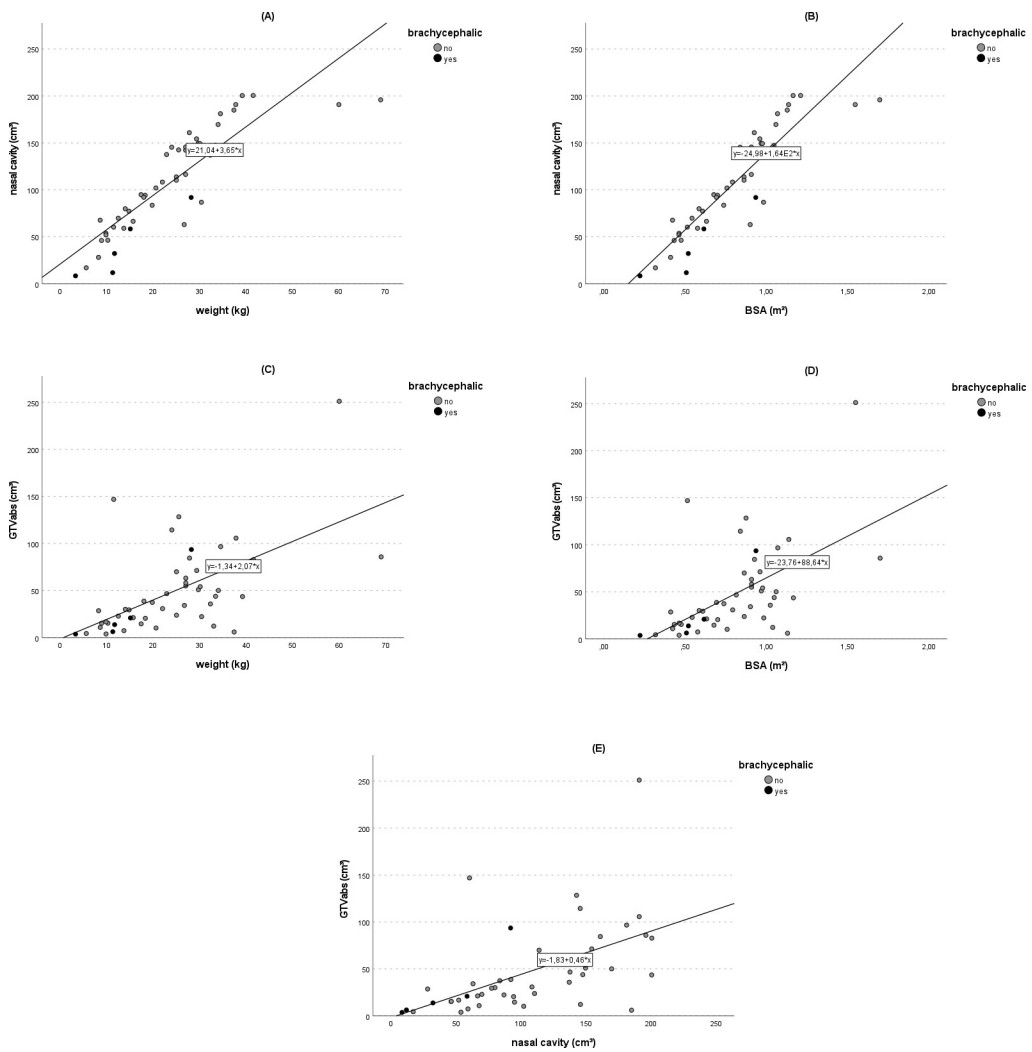

**Fig 1. Correlation between nasal cavity, tumor volume and dogs' sizes.** (A) The size of the nasal cavity (cm$^3$) was positively correlated with weight ($r$ = 0.87; p<0.001) and (B) BSA ($r$ = 0.90; p<0.001) in all 49 dogs. (C-E) The absolute GTV (GTVabs) also correlated moderately with dogs' weight, BSA, and with nasal cavity size (p<0.001). The black dots depict the five dogs of brachycephalic breeds.

## Correlation between nasal cavity, tumor volume and dogs' sizes

The size of the nasal cavity was positively correlated with dogs' body sizes (weight: $r$ = 0.87; p<0.001; BSA: $r$ = 0.90; p<0.001) (Fig 1A and 1B). The 5/49 (10.2%) brachycephalic dogs were not obvious outliers (Fig 1). Absolute GTV correlated moderately with dogs' weight (r = 0.59; p<0.001) and BSA (r = 0.58; p<0.001), as well as with the size of the nasal cavity (r = 0.55; p<0.001) (Fig 1C–1E). The correlations between the three measures of relative GTV and the three measures of dogs' sizes (nine correlations) ranged from r = 0.00 to r = 0.26 and were not statistically significant (S2 Table).

## Correlation between tumor volume and stage

There was no significant difference among patients with tumor stage 1 to stage 4 for absolute GTV (p = 0.28), GTV relative to weight (p = 0.09) and GTV relative to BSA (p = 0.13). For GTVs relative to nasal cavity, however, there was a statistically significant difference between

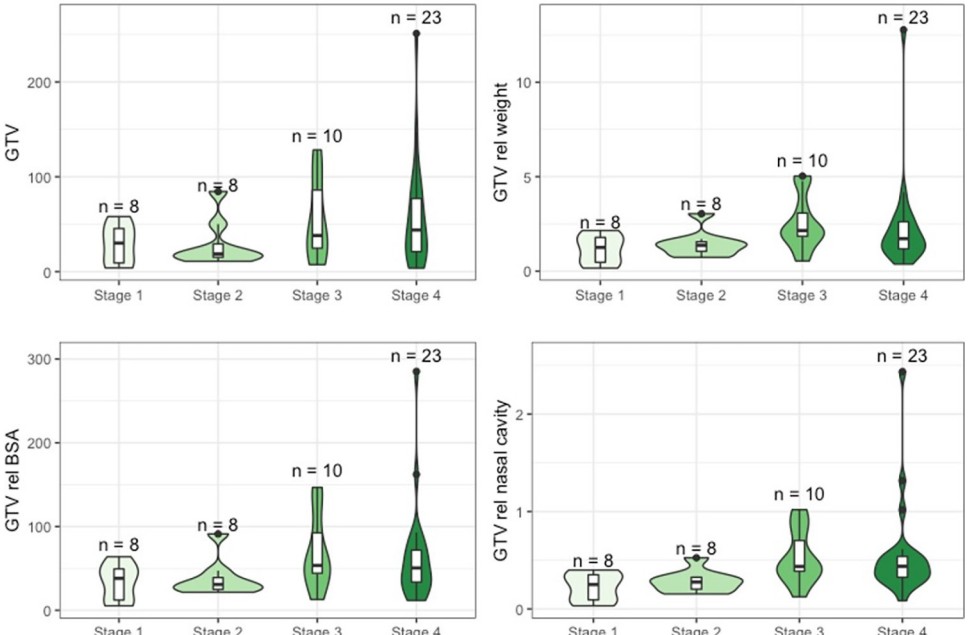

**Fig 2. Distribution of GTVs according to modified Adams tumor stage.** Violin plots depicting the distribution of absolute and relative tumor volumes of all 49 dogs depending on the different tumor stages. There was a statistically significant difference between GTVs relative to nasal cavity and the different tumor stages (p = 0.005). Here, stage 1 tumors were significantly smaller than stage 3 tumors (p = 0.026), stage 1 tumors smaller than stage 4 tumors (p = 0.026), stage 2 tumors smaller than stage 3 tumors (p = 0.04) and stage 2 tumors smaller than stage 4 tumors (p = 0.027).

different tumor stages (p = 0.005): when the GTV was corrected for the different nasal cavity sizes, stage 1 tumors were significantly smaller than stage 3 tumors (p = 0.026), stage 1 tumors smaller than stage 4 tumors (p = 0.026), stage 2 tumors smaller than stage 3 tumors (p = 0.04) and stage 2 tumors smaller than stage 4 tumors (p = 0.027) (Fig 2).

## Influence of protocol and tumor stage on outcome

Results of Kaplan-Meier analysis for TTP are shown in Fig 3. The two treatment groups did not differ in baseline parameters such as age, sex, weight, BSA, nasal cavity size or tumor

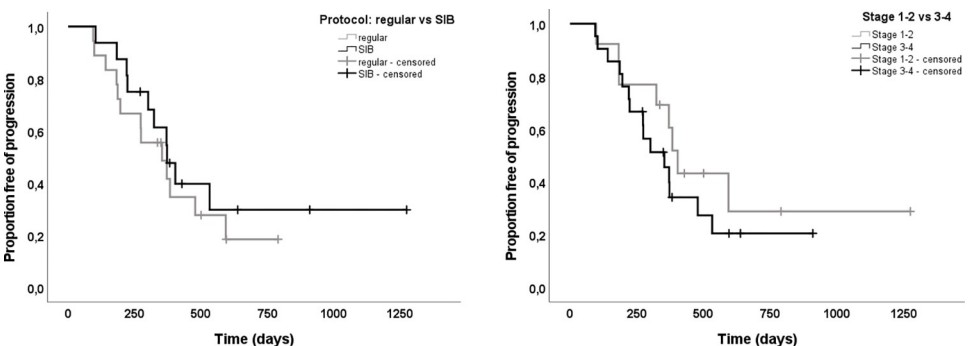

**Fig 3. Impact of radiation protocol and tumor stage on time to progression (TTP).** Kaplan-Meier plotting, and log-rank test showed no significant difference in TTP between (A) dogs with different radiation protocols (regular (10x4.2 Gy) versus SIB (GTV boosted to 10x4.83 Gy); p = 0.45), and (B) between dogs with different tumor stages (stage 1–2 versus stage 3–4; p = 0.37). Censoring is indicated by tick marks.

**Table 2. Impact of different tumor measures on TTP.**

| Tumor measures | Median [IQR] | Hazard ratio (95%CI) | P-value |
|---|---|---|---|
| GTVabs (cm$^3$) | 31.80 [15.15, 57.10] | 1.01 (0.99–1.02) | 0.19 |
| GTVrel_W | 1.43 [1.11, 2.11] | 1.37 (0.91–2.07) | 0.13 |
| GTVrel_BSA | 39.90 [25.90, 67.22] | 1.01 (0.99–1.02) | 0.14 |
| GTVrel_NC | 0.35 [0.26, 0.46] | 10.97 (1.25–96.06) | 0.03* |

GTVabs: Absolute tumor volume; GTVrel_W: Tumor volume relative to weight; GTVrel_BSA: Tumor volume relative to body surface area; GTVrel_NC: Tumor volume relative to nasal cavity; IQR: Interquartile range; CI: Confidence interval; *: Statistically significant.

volumes (absolute and relative) as shown in Table 1. Median TTP for patients with the regular protocol was 353 days (95%CI:187–520) and 372 days (95%CI:279–466) for patients with the SIB protocol. There was no statistically significant difference between protocols (p = 0.45) and further evaluation regarding influence of GTVs and tumor stage was therefore performed with all dogs together. Since there was a previously mentioned statistically significant difference between different tumor stages and GTVs relative to nasal cavity, we compared the results of TTP for the following both groups: stage 1–2 versus stage 3–4. Median TTP for patients with stage 1–2 was 403 days (95%CI:350–457) and 353 days (95%CI:224–483) for patients with stage 3–4, n.s. (p = 0.37).

## Influence of tumor measures on outcome

Time to progression was not influenced by absolute GTV (p = 0.19), GTV relative to weight (p = 0.13) and GTV relative to BSA (p = 0.14). However, there was a significant association between GTV relative to nasal cavity and TTP (p = 0.03): GTV relative to nasal cavity showed a hazard ratio of 10.97 (95%CI:1.25–96.06) for progression.

In addition, after adjusting for tumor stage (stage 1–2 versus stage 3–4) and treatment protocol, GTV relative to nasal cavity was still significantly associated with TTP (p = 0.04) and showed an increased hazard ratio of 11.08 (95%CI:1.07–114.89) for progression. The different tumor volume measures and influence on TTP are shown in Table 2.

**Table 3. Best tumor volume reduction and outcome.**

| | Regular protocol (n = 18) | SIB protocol (n = 16) | All (n = 34) |
|---|---|---|---|
| **Best tumor volume reduction (%)** median (range) | -2.27% (-91.78% to +216.99%) | -22.52% (-96.56% to +94.87%) | -11.54% (-96.56% to +216.99%) |
| **CR** n = (%) | 0 (0.0%) | 0 (0.0%) | 0 (0.0%) |
| **PD** n = (%) | 0 (0.0%) | 3 (18.8%) | 3 (8.8%) |
| **PR** n = (%) | 8 (44.4%) | 8 (50%) | 16 (47.1%) |
| **SD** n = (%) | 10 (55.6%) | 5 (31.3%) | 15 (44.1%) |

CR: Complete response; PD: Progressive disease; PR: Partial response; SD: Stable disease.

## Response assessment

The number of re-check CTs performed was as follows: One re-check CT was performed in sixteen dogs, two in ten dogs, three in four dogs, four in three dogs and five in one dog. Twelve dogs were imaged at the recommended time points 6 and 12 months after radiation therapy (and if still alive every 6 months thereafter). Of those dogs re-imaged at the recommended time points, 2/12 showed no clinical nasal signs but (only) clear disease progression according to CT 12 and 18 months after radiation therapy, respectively. Ten dogs were imaged at the defined time points but also showed mild to moderate clinical nasal signs. Twenty- two dogs were imaged due to suspected nasal tumor recurrence at other time points than the recommended ones. Dogs treated with the SIB protocol had a median best response of -22.52% (range -96.56% to 94.87%) tumor volume change, while dogs treated with the regular protocol had a median best response of -2.27% (range -98.95% to 216.99%) tumor volume change (Table 3). Sixteen dogs showed partial response and fifteen dogs showed stable disease as best response during the study period. Three dogs in the SIB group showed progressive disease at the time of re-imaging, i.e., no response but an increase in volume was visible at the time point of imaging. Most dogs eventually showed progressive disease during the time of the study (23/ 34; 68%) (Fig 4).

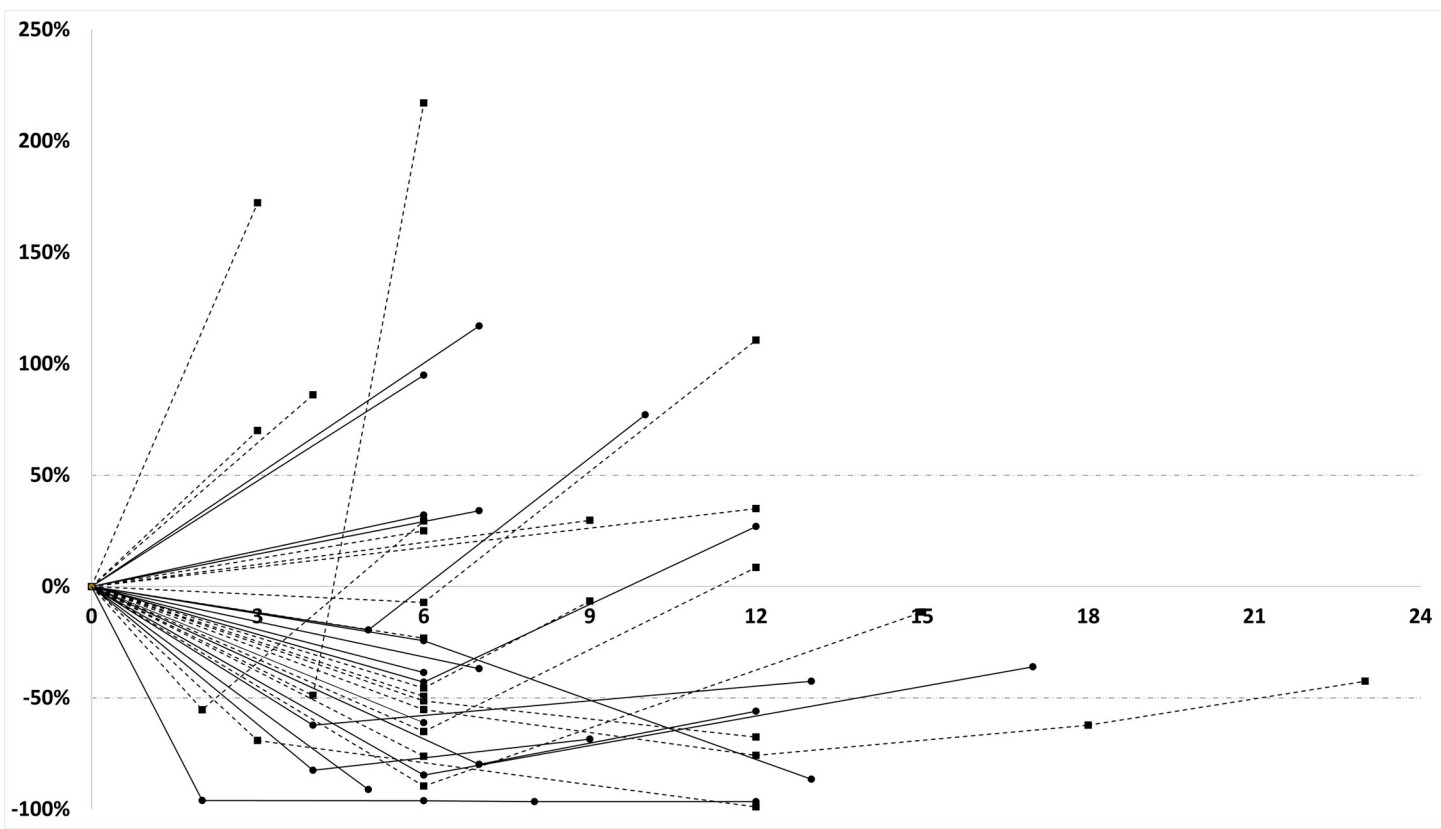

**Fig 4. Spider plots of tumor volume development after radiation therapy in 34 dogs.** Absolute GTV therapy with either the regular or the SIB protocol was termed "time 0 GTV". All other GTVs during follow-up were compared to this GTV or the smallest re-check GTV. Each line depicts a dog, each point or square is an imaging GTV at different time points. The dashed black line represents the absolute tumor volume of dogs irradiated with the regular protocol; the solid black line represents the absolute tumor volume of dogs treated with the SIB protocol.

## Discussion

In the study presented herein, we evaluated tumor volume as a prognostic factor in a heterogeneous dog population with sinonasal tumors. Two different types of tumor volume (absolute versus relative) were compared and also correlated with tumor stage. As hypothesized, GTV relative to nasal cavity showed an increased risk of progression in dogs with relatively larger tumors. Also, when looking at GTV relative to nasal cavity, stage 3 and 4 tumors were larger than stage 1 and 2 tumors.

The importance of prognostic factors is controversially discussed [14], especially regarding tumor volume and stage. In addition, GTV as a main variable and its effect on outcome has been described in few studies and the discrepancy to our current study should be mentioned: Poirier et al. (2021) and Bradshaw et al. (2015) used GTV as variable more or less than the median and evaluated the association with progression- free survival and/or overall survival [3,7]. Morgan et al. (2018) detected that the tumor volume decrease in relation to nasal cavity was associated with overall survival [1]. Nell et al. (2020) discussed—even if the GTV was not a main objective—that the lack of demonstrable prognostic impact of tumor volume may be related to variability in tumor size [2]. Namely, there was a trend that a smaller initial tumor volume led to a better outcome, but if a certain tumor volume was reached, this effect was lost. Additionally, a feasible dependency of tumor volume and dog conformation and size was mentioned. This point of unclarity could possibly be eliminated with our relative GTV, which takes the different dog sizes into account.

Considering the potential prognostic impact of tumor volume, we were interested in the question whether it is valid to compare the initial tumor volume in a large dog's nose with the one in a small dog's nose. As to be expected, we found that larger dogs have larger nasal cavities and, accordingly, larger absolute tumor volumes. But this positive correlation does no longer apply to the relative tumor volume. Hence, smaller dogs have absolutely, but not relatively smaller tumors.

The current dissent about prognostic factors in sinonasal tumors in dogs is also visible with different tumor stages: there are several reports stating that higher tumor stage is prognostically unfavourable [4–6], but other studies saw no effect of different tumor stages on progression- free interval, progression- free survival or overall survival [2,3,7].

In our study, we did not find any difference in TTP among the tumor stages when comparing stage 1–2 versus stage 3–4. Thus, higher tumor stages were not associated with a worse outcome in our dog population. Stevens et al. (2020) suggested that improved outcome for stage 4 tumors might be due to improved tumor coverage in the region of the cribriform plate with IMRT, in comparison to earlier, less conformal techniques [15].

Furthermore, different tumor stages showed a significant correlation with GTVs relative to nasal cavity. Hence, dogs with stage 3 and 4 have indeed larger tumor volumes if corrected for the nasal cavity sizes.

In our study, tumor volume relative to nasal cavity showed a hazard ratio of 10.97 for progression, i.e., every unit of increase in GTV relative to nasal cavity is correlated with an increase in hazard by 10.97. For example, if we compare two patients–dog one with GTV relative to nasal cavity of 0.4 and dog two with GTV relative to nasal cavity of 0.1 –our model estimates that dog one would have 0.3 x 10.97 = 3.29 times the hazard of dog two. Higher hazard ratio corresponds to an increased risk of progression, even after adjusting for tumor stage and treatment protocol. In contrast to this finding, a significant association between absolute tumor volume and TTP could not be detected. We would therefore recommend considering body size, e.g., correction for nasal cavity size when assessing tumor volume as a prognostic parameter in sinonasal tumors in dogs. Thus, large tumor volumes relative to nasal cavity

rather than large absolute tumor volumes should be considered as important risk factor of developing progression.

Several different radiation protocols are described. However, optimal radiation dose has not yet been found [16]. The inconsistency in methodology particularly in treatment protocols and margins impedes comparison between different reports [14]. Even results of similar protocols might vary. Whereas a theoretical study predicted an increased tumor control probability with a boost approach applied in 10 fractions [17], an earlier study was not able to see a positive effect of the boost protocol on TTP [7]. Also in a recent (although not randomized) study we were not able to detect a benefit of increased dose (via SIB) in dogs with sinonasal tumors [8]. While there is no randomized prospective clinical trial evaluating different treatment protocols in sinonasal tumors in dogs, most resulted in similar median survival times of 6.7 to 19.7 months after definitive-intent, finely fractionated radiation therapy [4,16]. Thus, attempts to predict TTP in canine sinonasal tumors after radiation therapy remain challenging and dose escalation or different approaches should be evaluated further.

Only few studies keep TTP in focus [18,19] and prognostic significance of several factors often refers to progression- free survival and overall survival but not to TTP. Survival time is influenced by other factors than tumor-related death alone and in addition we looked at progression based on imaging. Since we could recognize first indications of a possible correlation between relative tumor volume and outcome (TTP), we therefore decided to investigate TTP. In our cohort of dogs, we could not see a significant difference in TTP between the two different radiation protocols (regular versus SIB), even though dogs treated with the SIB protocol showed a stronger median best response in tumor volume reduction compared to the regular protocol (-22.52 versus -2.27%). But this might have depended on better follow-up with imaging (routine re-checks, not only in case of clinical nasal signs). In our study, median TTP was 353 days for patients with the regular protocol, which is comparable to the TTP of Galloway et al. (2020) with 335 days (95% CI 264–544 days) and Thrall et al. (1993) with 312 days [18,19]. For our patients with SIB protocol, median TTP was not different with 372 days.

In the present study, tumor volumetric response after several months showed that most tumors progress locally. An important reason for the clustered occurrence of progressive disease could be the inclusion criteria of having a re-check CT scan at any time point after the end of radiation therapy. Diagnostic imaging examinations are often executed when dogs showed clinical signs of progressive disease. Dogs without evidence of clinical nasal signs might therefore not have been included, if the dog owner did not elect a re-check CT. Because imaging was not performed at defined time points in all dogs, it is unclear if dogs showed better volume response to treatment between the end of radiation therapy and the re-check CT.

Several limitations need to be noted: 1) The study is retrospective: therefore, the authors rely on information available in the medical records. 2.) The small number of subjects in our sample may mean that we did not have enough statistical power to exclude a type II error (false negative). Since TTP is based on a small group of dogs, prognostic relevance of absolute or relative GTV and tumor stage could shift if a large population were examined. However, our intention was to start a discussion about relative, rather than absolute GTV as a possible prognostic factor in the future; further research is certainly needed. 3.) Because a heterogeneous group of dogs was included, other factors such as tumor type and differentiation could also have influenced outcome. Especially the decision to include two dogs with benign tumors and three with unknown histology could potentially have influenced outcome. 4.) Because clinical response is subjective and clinical progressive disease is difficult to differentiate from chronic rhinitis, we decided to assess the response according to imaging only. However, the lack of uniform follow-up regarding time-points also does not allow for an accurate tumor response comparison. Time points of re-check CTs varied: while some were routinely planned

6 and 12 months after radiation therapy (and some detected early, symptom-free failures), others were performed due to recurrent clinical signs and/or suspected tumor relapse at various time points. Therefore, given that not all patients were imaged and that the time of imaging varied, our results need to be interpreted with caution.

## Conclusion

In conclusion, individual patient size should be included when assessing prognostic influence of tumor volumes in dogs with sinonasal tumors. We confirmed our hypothesis that *relative* rather than absolute GTV may influence TTP in our cohort of dogs. Also, larger tumor volume relative to nasal cavity was correlated with higher tumor stage. While our study population was small and heterogeneous, we want to stimulate the idea, that GTV *relative to nasal cavity* should be evaluated as a prognostic factor in future studies.

## Supporting information

**S1 Table. Patient and tumor characteristics for part one (n = 49).** GTVabs: Absolute tumor volume; GTVrel_W: Tumor volume relative to weight; GTVrel_BSA: Tumor volume relative to body surface area; GTVrel_NC: Tumor volume relative to nasal cavity; IQR: Interquartile range; SD: Standard deviation.
(PDF)

**S2 Table. Correlations between relative GTVs and dogs' sizes.** GTVrel_W: Tumor volume relative to weight; GTVrel_BSA: Tumor volume relative to body surface area; GTVrel_NC: Tumor volume relative to nasal cavity.
(PDF)

## Acknowledgments

The authors thank the statistical experts Sandar Felicity Lim (MSc., currently working as Data Scientist for the Expedia Group in Geneva, Switzerland) and Prof. Dr. Richard Evans of the Iowa State University for the support with statistical analysis.

## Author Contributions

**Conceptualization:** Felicitas Czichon, Carla Rohrer Bley, Valeria Meier.

**Data curation:** Felicitas Czichon, Valeria Meier.

**Formal analysis:** Felicitas Czichon, Carla Rohrer Bley, Valeria Meier.

**Funding acquisition:** Carla Rohrer Bley.

**Investigation:** Felicitas Czichon, Carla Rohrer Bley, Valeria Meier.

**Methodology:** Felicitas Czichon, Carla Rohrer Bley, Valeria Meier.

**Project administration:** Felicitas Czichon, Carla Rohrer Bley, Valeria Meier.

**Resources:** Carla Rohrer Bley.

**Software:** Carla Rohrer Bley, Valeria Meier.

**Supervision:** Carla Rohrer Bley, Valeria Meier.

**Validation:** Carla Rohrer Bley, Valeria Meier.

**Visualization:** Felicitas Czichon, Carla Rohrer Bley, Valeria Meier.

**Writing – original draft:** Felicitas Czichon, Valeria Meier.

**Writing – review & editing:** Felicitas Czichon, Carla Rohrer Bley, Valeria Meier.

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
