## [Decision Letter · Decision Letter 0]

19 Jan 2022

PONE-D-21-40338Do we need size correction for assessing tumor volume as a prognostic parameter in sinonasal tumors in dogs?PLOS ONE

Dear Dr. Czichon,

Thank you for submitting your manuscript to PLOS ONE. After careful consideration, we feel that it has merit but does not fully meet PLOS ONE’s publication criteria as it currently stands. Therefore, we invite you to submit a revised version of the manuscript that addresses the points raised during the review process.

Please address all reviewer comments.

We look forward to receiving your revised manuscript.

Kind regards,

Douglas H. Thamm, V.M.D.

Academic Editor

PLOS ONE

Journal Requirements:

"This work was supported by the Swiss National Science Foundation (SNSF), http://www.snf.ch/en, grant number: 320030-182490 (PI: Carla Rohrer Bley). The funders had no role in study design, data collection and analysis."

"This work was supported by the Swiss National Science Foundation (SNSF), http://www.snf.ch/en, grant number: 320030-182490 (PI: Carla Rohrer Bley). The funders had no role in study design, data collection and analysis, decision to publish, or preparation of the manuscript."

Reviewers' comments:

Reviewer's Responses to Questions

**Comments to the Author**

1. Is the manuscript technically sound, and do the data support the conclusions?

Reviewer #1: Partly

Reviewer #2: Yes

2. Has the statistical analysis been performed appropriately and rigorously? 

Reviewer #1: No

Reviewer #2: Yes

3. Have the authors made all data underlying the findings in their manuscript fully available?

Reviewer #1: No

Reviewer #2: Yes

4. Is the manuscript presented in an intelligible fashion and written in standard English?

Reviewer #1: No

Reviewer #2: Yes

5. Review Comments to the Author

Reviewer #1: Review of Do we need size correction for assessing tumor volume as a prognostic parameter in sinonasal tumors in dogs?

This is a retrospective study in dogs with sinonasal tumor treated with 2 radiation protocols where the author wants to determine if the GTV either absolute or relative can be use for prognosis of outcome.

Overall, the primary research question as in is relative tumor size a more informative prognostic indicator than absolute tumor size is valid. However, there is major limitations to the study in its current form to answer the research question.

Some of the stated objectives of their study like pattern of failure does not articulate well in the paper and they do not make its relevance clear. Also although not stated in their objective, the emphasis placed on the comparison between radiation protocol is not warranted as it stands.

In general the study is hard to read as the wording is imprecise and does not flow. The primary outcome of time to progression is not appropriate since less than half of the dogs (23/49 (47%)) had documented imaging progression and only a bit more than 2/3 of dogs actually had imaging follow-up. The author did not define what to them is progression (i.e. imaging, clinical signs) and do not provide us with adequate follow-up information timing of CTs, recommended follow-up, which clinical signs they used to determine progression etc… Since the cohort of dogs were presented at a conference 15 months ago, I would assume that the data is mature and would highly suggest that the authors uses overall survival and/or progression-free survival as a primary outcome rather than time to progression that is more biased.

We used the Strobe cohort study reporting guidelines to evaluate the paper: https://www.strobe-statement.org/checklists/ and also used the consort: http://www.consort-statement.org/ one as some of the details are almost more clinical trial versus observational study

Item1

Title: the title does not report the study design nor does it report the treatment that the dogs received. It lacks clarity. Authors need to offer a new one.

Abstract. Sentences and words are awkward exemple line 39-41 I cannot really understand the meaning of the sentence. Please see Strobe conference abstract for details to be included in the abstract.

Introduction

Item2

There is no rational or background for their second objective-pattern of failure in the introduction. I would suggest to remove the second part or somehow address it into a cohesive study design and research question.

For the GTV-

Reference study 1 and 2 both uses PFS as primary outcome and 1 uses OS also as primary outcome. Both uses GTV as a dichotomous variable as less or more than the median GTV which was not performed in the current study.

Reference 3 does not report GTV and do not evaluate GTV with outcome.

Reference 4 did specify that outcome (they used progression free-interval and eliminated dogs that did not have a 3 months recheck CT as they could not reliably evaluate the outcome with only a 6 months follow-up CT) was not the objective of their study and they concluded that:” specifically,therewas a trend for all investigators in that smaller the baseline tumorvolume, the better the outcome; however, once a particular tumorvolume was reached, this effect was lost (Figure7A).” So possibly would have been found prognostic if GTV had been measured as a dichotomous variable.

Reference 5 did use GTV relative to nasal size volume similar to the present study and not absolute GTV. They used overall survival as primary outcome.

So this illustrate that your primary research question is a great one but you should make the reader aware of the discrepancy in the literature as it comes to GTV measurement and primary outcome. And then design your study to more adequately answer the question.

Line 58-wording is strange what is an inspected factor? You want to introduce the idea of stage as it might correlate with your primary variable which is GTV. This is unclearly done.

Line 68-69 not sure what this sentence means. Do you mean that since dogs have different body size/weight/nasal cavity measurement, you need to find a way to take this into account and modify the GTV measurement as needed and this is what you will do in this research project.

Now you should state your hypothesis as it regards to GTV and leave the protocol/stage/pattern of failure for the material and methods.

I would remove the pattern of failure as one of your aim and hypothesis as it is a completely different unrelated research question.

Methods

Item 4-Please specify study design-like single institution retrospective cohort study

Item 5=Setting-please specify all the details of the study here as per Strobe. Location/dates/follow-up/data collection etc…. You cannot just refer all to a conference abstract it needs to be here in the paper

Item 6-give how dogs were selected, how they were diagnosed , how dogs were allocated to the treatment protocol. Inclusion and exclusion criteria as to me you should only include dogs with histologically confirmed sinonasal tumor (maybe also the cytological one maybe) and exclude the benign tumors and possibly the zebra (i.e. nasal lymphoma/melanoma etccc) so you have a representative sample of typical sinonasal tumor that your can have external validity and with enough info that your reader can evaluate.

You should provide as much information on the histology of your cohort such as the sarcoma: how many were chondrosarcoma versus osteosarcoma versus fibrosarcoma, for the carcinoma-who was adenocarcinoma/squamous cell carcinoma versus undiffererentiated etc… And once again, the unknowns, benign and potentially others should not be included.

Although the RT description is not the primary aim, you need to provide the details of dose prescription, GTV/CTV/PTV contouring. Who does the contouring etc…. Consider since there is no words limits to report everything required by

Keyerleber MA, McEntee MC, Farrelly J, Podgorsak M. Completeness of reporting of radiation therapy planning, dose, and delivery in veterinary radiation oncology manuscripts from 2005 to 2010. Vet Radiol Ultrasound. 2012 Mar-Apr;53(2):221-30.

Primary outcome needs to be well defined and ideally would be overall survival and/or progression-free survival. Please define how progression was determined if you will use progression-free survival as it can come from imaging but also clinical signs etc…. As it stands, less than 50% of your cohort had progression based on imaging.

Please define clearly how the initial GTV was contoured in which software and how it was defined. Was it contoured by only 1 rater. Mutliple rater, only once? Cannot refer to a conference abstract, details need to be in the current paper.

Please clearly define the recommended follow-up in the methods and the follow-up achieved in the results.

Since I feel that the primary outcome needs to change and that the inclusion and exclusion criteria needs to be specified prior to even evaluate the results at this point.

Reviewer #2: The manuscript entitled "Do we need size correction for assessing tumor volume as a prognostic parameter in sinonasal tumors in dogs? " describes a study that looks at the effect of adjusted tumor size on outcomes. Overall this is interesting and while this study does not definitively answer the question proposed it does raise interesting points for further evaluation.

I have a few minor comments:

Under the response and pattern of failure section (lines 253-268) it would be useful to the reader to present at least summary statistics regarding number of CT scans per dog evaluated, time to evaluation, how many were evaluated because of suspicion of progression or recurrence?

While the English overall is quite good there are multiple points where it is a bit awkward. For example line 282 "Herein, we revised" do you mean We evaluated? Perhaps consider having a native English language speaker review the manuscript.

6. PLOS authors have the option to publish the peer review history of their article (what does this mean?). If published, this will include your full peer review and any attached files.

Reviewer #1: No

Reviewer #2: No

---

## [Author Response · Author response to Decision Letter 0]

4 Mar 2022

Dear Journal Editorial Team,

Dear Reviewers,

first of all, we would like to thank you for your time and effort to revise our manuscript. Your constructive feedback helped us to make individual paper sections more precise and fluent. Thus, we have adjusted the manuscript according to the suggested changes and would like to resubmit it for further review. We sincerely hope that you approve of our changes. 

Please find the detailed response to all the reviewers' comments in the file "Response to Reviewers". 

Sincerely, 

Felicitas Czichon

---

## [Decision Letter · Decision Letter 1]

29 Mar 2022

PONE-D-21-40338R1Relative tumor volume has prognostic relevance in canine sinonasal tumors treated with radiation therapy: A retrospective studyPLOS ONE

Dear Dr. Czichon,

Thank you for submitting your manuscript to PLOS ONE. After careful consideration, we feel that it has merit but does not fully meet PLOS ONE’s publication criteria as it currently stands. Therefore, we invite you to submit a revised version of the manuscript that addresses the points raised during the review process.

I agree with the opinion of Reviewer 1 that progression free interval / progression free survival is an inappropriate endpoint for this cohort of patients, given that not all patients were imaged and that the time of imaging varied. I agree that overall survival time is a more appropriate endpoint. Please adjust accordingly. ==============================

We look forward to receiving your revised manuscript.

Kind regards,

Douglas H. Thamm, V.M.D.

Academic Editor

PLOS ONE

Reviewers' comments:

Reviewer's Responses to Questions

**Comments to the Author**

1. If the authors have adequately addressed your comments raised in a previous round of review and you feel that this manuscript is now acceptable for publication, you may indicate that here to bypass the “Comments to the Author” section, enter your conflict of interest statement in the “Confidential to Editor” section, and submit your "Accept" recommendation.

Reviewer #1: (No Response)

Reviewer #2: All comments have been addressed

2. Is the manuscript technically sound, and do the data support the conclusions?

Reviewer #1: Partly

Reviewer #2: Yes

3. Has the statistical analysis been performed appropriately and rigorously? 

Reviewer #1: No

Reviewer #2: Yes

4. Have the authors made all data underlying the findings in their manuscript fully available?

Reviewer #1: Yes

Reviewer #2: Yes

5. Is the manuscript presented in an intelligible fashion and written in standard English?

Reviewer #1: Yes

Reviewer #2: Yes

6. Review Comments to the Author

Reviewer #1: Good work on improving the manuscript. Some of my major concerns have not been addressed-see attached review.

Reviewer #2: Thank you for the response to my comments and I feel that the authors have addressed my concerns.

A few small suggestions:

Line 137 change 10,1 to 10.1

I am surprised that age and weight were normally distributed as this is often not the case. This is suggested since they are presented as means and methods state normally distributed data is presented this way. Would suggest confirming this.

Line 168 - you state data for medians are presented as "median (95% confidence interval (95% CI) or range)" in tables you present with IQR - this is fine in my opinion but please add this to info in statistical section.

7. PLOS authors have the option to publish the peer review history of their article (what does this mean?). If published, this will include your full peer review and any attached files.

Reviewer #1: No

Reviewer #2: **Yes: **Michael S Kent

---

## [Author Response · Author response to Decision Letter 1]

14 Apr 2022

Dear Journal Editorial Team, 

Dear Reviewers,

thank you for your time and effort to provide constructive feedback on our manuscript. We have reflected on the feedback and incorporated your suggestions into our manuscript. In our opinion, it has gained more clarity and quality as a result. We hereby resubmit it for further review and sincerely hope that you approve of our changes.

Please find the detailed response in the document "Response to Reviewers".

Sincerely, 

the authors

---

## [Decision Letter · Decision Letter 2]

16 May 2022

Relative tumor volume has prognostic relevance in canine sinonasal tumors treated with radiation therapy: A retrospective study

PONE-D-21-40338R2

Dear Dr. Czichon,

We’re pleased to inform you that your manuscript has been judged scientifically suitable for publication and will be formally accepted for publication once it meets all outstanding technical requirements.

Kind regards,

Douglas H. Thamm, V.M.D.

Academic Editor

PLOS ONE

Additional Editor Comments (optional):

Reviewers' comments:

Reviewer's Responses to Questions

**Comments to the Author**

1. If the authors have adequately addressed your comments raised in a previous round of review and you feel that this manuscript is now acceptable for publication, you may indicate that here to bypass the “Comments to the Author” section, enter your conflict of interest statement in the “Confidential to Editor” section, and submit your "Accept" recommendation.

Reviewer #1: All comments have been addressed

Reviewer #2: All comments have been addressed

2. Is the manuscript technically sound, and do the data support the conclusions?

Reviewer #1: Partly

Reviewer #2: Yes

3. Has the statistical analysis been performed appropriately and rigorously? 

Reviewer #1: I Don't Know

Reviewer #2: Yes

4. Have the authors made all data underlying the findings in their manuscript fully available?

Reviewer #1: Yes

Reviewer #2: Yes

5. Is the manuscript presented in an intelligible fashion and written in standard English?

Reviewer #1: Yes

Reviewer #2: Yes

6. Review Comments to the Author

Reviewer #1: I think with the very clear limitations outlined in the discussion, the paper is acceptable in his corrected form. Good job.

Reviewer #2: Thank you for addressing my concerns. I feel you have addressed the limitations of the study and tempered your conclusions.

7. PLOS authors have the option to publish the peer review history of their article (what does this mean?). If published, this will include your full peer review and any attached files.

Reviewer #1: No

Reviewer #2: **Yes: **Michael S Kent

---

## [Editor Report · Acceptance letter]

20 May 2022

PONE-D-21-40338R2 

Relative tumor volume has prognostic relevance in canine sinonasal tumors treated with radiation therapy: A retrospective study 

Dear Dr. Czichon:

I'm pleased to inform you that your manuscript has been deemed suitable for publication in PLOS ONE. Congratulations! Your manuscript is now with our production department. 

Kind regards, 

on behalf of

Dr. Douglas H. Thamm 

Academic Editor

PLOS ONE